# Are UK Rivers Getting Saltier and More Alkaline?

**Shan Jiang \*, Xuan Wu, Sichan Du, Qin Wang and Dawei Han**

Department of Civil Engineering, University of Bristol, Bristol BS8 1TR, UK
\* Correspondence: og20481@alumni.bristol.ac.uk; Tel.: +86-191-5105-1570

**Abstract:** River salinisation and alkalinisation have become one of the major environmental problems threatening the safety of global freshwater resources. With the accelerated climate change and aggravating anthropogenic influences, it is important to identify the trends and causes of river salinisation and alkalinisation so that better mitigation measures could be taken. This study has focused on the UK rivers because there has been insufficient investigation on this topic. To understand the salinisation and alkalinisation trends and causes of rivers in the UK over the past 20 years from a vertical (analysis of each river) and horizontal (comparison of all rivers) perspective, this study uses the Theil-Sen regression and Mann-Kendall test to deal with the trends of conductivity (proxy on salinisation) and pH (proxy on alkalinisation), obtains outliers of conductivity and pH by boxplot, and calculates the Pearson's and the Kendall's Tau correlation coefficients ($\alpha = 0.05$) between the water quality data and the potential factors (potential road salting, normalized difference vegetation index (NDVI), river discharge, agricultural and urban lands). The results show that the UK rivers are becoming more alkaline with a median pH increase of 0.05 to 0.40, but less salty with a median conductivity decrease of 0.06 to 0.11 mS/cm. And the changes in conductivity and pH have seasonality and regionality, which shows that there are usually greater changes in trends and medians of them in winter or through reaches with more anthropogenic disturbance. Furthermore, from a vertical perspective, the conductivity of more than 50% of rivers in this study is negatively correlated with NDVI and river discharge, and positively correlated with potential road salting, and the pH of that is positively correlated with agricultural lands. While from a horizontal perspective, NDVI and agricultural lands are positively correlated with pH, and potential road salting and urban lands are positively correlated with conductivity. Therefore, road salting, urbanisation, agricultural lands, river discharge and vegetation cover can be considered to affect river salinisation and alkalinisation in the UK.

**Keywords:** UK rivers; salinisation; alkalinisation; pH; conductivity

## 1. Introduction

River salinisation and alkalinisation are becoming one of the important factors affecting the global river ecosystem [1], which leads to the decline of biodiversity, the degradation of agricultural lands and freshwater areas, and the corrosion of infrastructure, thus threatening human living conditions [2–4]. Studying the changes in river salinisation and alkalinisation and analysing their causes will help to better protect freshwater resources and cope with the shortage of freshwater resources.

Generally, river salinisation refers to the accumulation of ions, leading to an increase in conductivity or total dissolved solids [5], which can be caused by natural (e.g., climate change, weathering and seawater intrusion) or anthropogenic (e.g., road salt, mining, agriculture and groundwater abstraction) processes [6–8]. The research on river salinisation can be divided into two periods. Before the 21st century, most researchers regarded that the decline of soil permeability caused by agricultural activities made rivers tend to be salinised in semi-arid and arid areas [9,10]. However, recent studies show that the salinity of rivers in many humid regions is also increasing [11,12], which may be

attributed to the heavy salting of roads for winter de-icing (e.g., more extreme cold spells due to climate change), chemical weathering, mining, seawater inversion and agricultural activities [13]. Among them, the long-term use of road salt in North America leads to many inputs of $Na^+$ and $Cl^-$, which is one of the main reasons for salinisation aggravation in many rivers. For example, in the eastern USA, the concentration of $Na^+$ and $Cl^-$ in rivers near New York State has increased approximately twice in 50 years as a result of winter de-icing [14,15], and a large amount of $Ca^{2+}$ and $Mg^{2+}$ has been observed to be displaced from the soil by $Na^+$, and finally discharged into rivers through surface runoff, which intensifies the salinisation of rivers in these areas [16,17]; Similar phenomena have been observed in southeastern Canada, both in Ontario and Quebec with the highest winter road salting, resulting in varying degrees of salinity increases in their freshwater areas [18]. On the other hand, chemical weathering due to acidic deposition is the main cause of river salinisation in warm and humid areas of North America. For instance, Davies [19] and Kaushal [20] have found that urban impervious surfaces release $Ca^{2+}$ and $K^+$ under the influence of chemical weathering, thereby increasing the salinity of urban groundwater and surrounding watersheds. In addition, the salinity of the Rhine has been not only affected by potash mining, but threatened by seawater intrusion caused by climate warming [21,22]. In Australia, salinity in soils and rivers has also been increasing as saline groundwater levels rise with the increase in agricultural land use and fertilisers [23,24].

The alkalinisation of rivers refers to the increase of pH in the river, which is usually affected by temperature, aquatic biological activities, acidic deposition, chemical weathering, sewage discharge and land use, which is similar to the causes of river salinisation [1,13,25]. In the past, river alkalinisation was usually seen only as a recovery from acidification, which had not been studied much until the last 20 years. Currently, these phenomena are observed in many regions, such as the eastern USA and north-western France [26–28]. In the USA, there is a strong link between river salinisation and alkalinisation, which manifests as a simultaneous upward trend in conductivity, pH and base cations in the rivers. This phenomenon is called freshwater salinisation syndrome by Kaushal [13]. It may be explained as anthropogenic salt inputs, chemical weathering and agriculture, which promote base cation exchange in soil and rock, and the precipitated cations are channeled through surface runoff, thus altering the concentration of base cations as well as bicarbonates in the rivers and exacerbating salinisation and alkalinisation [22,29]. Its severity is also related to the hydrological and meteorological conditions of the study site [30,31].

However, despite the importance of the topic, there was relatively little research on river salinisation and alkalinisation in the UK, which was mainly carried out in the 1980s as a supplement to acid rain research [32]. As one of the countries with the highest urbanisation in Europe, it is important to study whether the UK rivers are threatened by salinisation and alkalinisation, which will not only help the UK better protect freshwater security, but also may be a significant reference for understanding the salinisation and alkalinisation of rivers in other European countries with a temperate climate and high urbanisation. This study is intended to test a hypothesis that the salinisation and alkalinisation of rivers in the UK has also shown an upward trend in recent years (e.g., over the last 20 years), as it has in other countries. To verify the hypothesis, this study selects a few representative river basins and collects the related water quality data on river salinisation and alkalinisation, then analyses the trends, similarities and differences of the relevant data. At the same time, this study also explores the possible causes and hazards of river salinisation and alkalinisation trends.



## 2. Materials and Methods

### 2.1. Study Sites

Considering the high level of urbanisation in the UK, to make the results more representative, better reflect the trends of river salinisation and alkalinisation in different parts of the UK and explore their causes, this study took into account factors such as the level of anthropogenic disturbance, types of landcover, climate, geological environment, and availability of water quality data, and selected the 50 Environmental Agency's Water Quality Data Archive (EAWQDA) freshwater monitoring sites of the River Tame, River Trent, River Mersey and River Avon (Figure 1) with differences in these factors as the research objects near the major cities in the UK. In addition, to gain a deeper understanding of the salinisation and alkalinisation of each river, 30 special sampling sites were selected from the 50 EAWQDA sites, which had full length data, closeness to roads as well as special areas and located in different river reaches. For instance, the special sampling sites in the midstream of the River Tame are at the entrance and outlet of Lake Lea Marston, which has a purifying effect on water quality [33]; The special sampling sites in the middle and lower River Mersey are near the entrance and exit of the river into the Manchester Canal, which is an important channel with serious pollution [34].

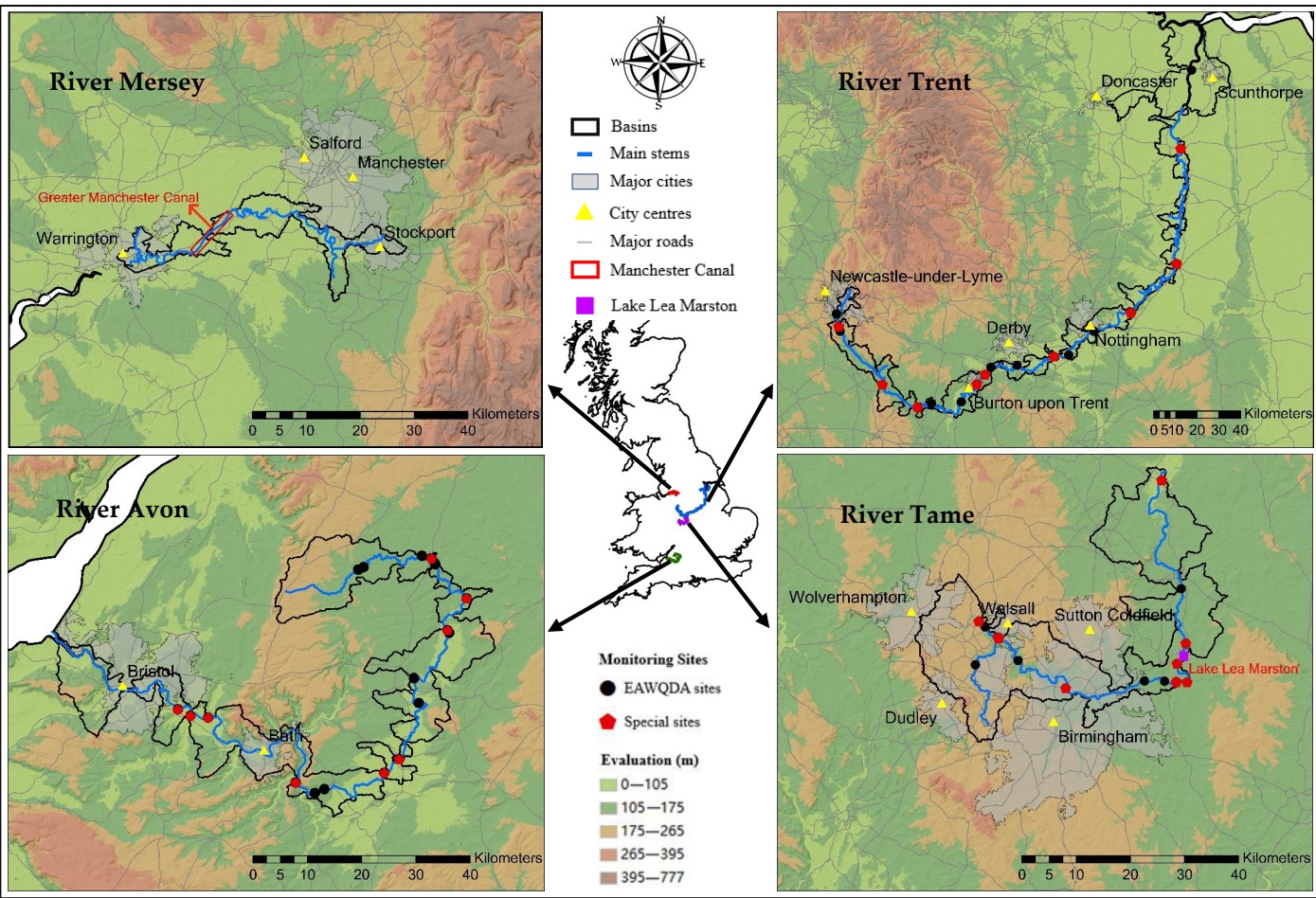

**Figure 1.** Main stems of the four rivers, with yellow triangles representing the main cities around the basins, grey lines for the major roads, black dots for the EAWQDA freshwater monitoring sites, and red pentagons for the special sampling sites.

*2.2. Data Collection*

The indicators used to represent river salinisation and alkalinisation mainly include pH, conductivity, base cations, dissolved oxygen (DO), dissolved inorganic carbon (DIC) and dissolved organic carbon (DOC), etc., [12,29]. However, only the monthly data of conductivity and pH were available from the 50 EAWQDA freshwater monitoring sites over a long period (at least 10 years). Thus, these two types of water quality data were selected for this study and used to represent river salinisation and alkalinisation, respectively.

As mentioned before, although there were many natural and anthropogenic factors affecting the river salinisation and alkalinisation, considering the availability of data, this study planned to use agricultural lands, urban lands, NDVI, river discharge and road salting data over the past 20 years to explore the causes of conductivity and pH trends in the four rivers. Among them, the annual data of agricultural and urban lands (km²), the 16-day data of NDVI, and the daily data of river discharge (m³/s) were obtained from the MCD12Q1, the MOD13A2 (https://search.asf.alaska.edu, accessed on 5 July 2021), and the National River Flow Archive (https://nrfa.ceh.ac.uk/, accessed on 16 May 2021), respectively. Unfortunately, there were no accurate records of road salting in the UK, so this study collected the daily air temperature, the annual total length of the classified roads (major and minor roads), the average width of the classified roads, usage of salting per square meter, and empirical difference of air-to-road temperature from CAMELS-GB [35], OS OpenData (https://osdatahub.os.uk/, accessed on 6 July 2021) and Met Office (https://wow.metoffice.gov.uk/, accessed on 13 June 2021) for each river, respectively, to simulate a new variable—the annual potential road salting (winter)—to replace the real road salting. The basis of this simulation includes: (1) The UK Highways Act 1980 stipulates that road salting is required in advance when there is a risk of freezing on the road; (2) The road freezing is related to road surface temperature (RST), which has about a 4 °C difference from the air temperature between October to April (https://www.metoffice.gov.uk/services/transport/road/opensite-resources, accessed on 15 June 2021); (3) If the RSTs are going to fall below 0.5 °C, the company responsible for salting will salt on the classified roads, including all major roads ('A' roads and motorways (M roads)) as well as some minor roads ('B' and 'C' roads), and the amount of salting is about 16 g/m² (https://www.buckinghamshire.gov.uk/parking-roads-and-transport/, accessed on 15 June 2021); (4) The usual width of the classified roads with one lane is 3.65 m (https://www.gov.uk/government/publications/uk-road-width-restrictions-foi, accessed on 15 June 2021). Its specific calculations are shown later.

*2.3. Data Pre-Processing*

Generally, water quality data have the characteristics of the irregular monitoring period, non-normality, measurement errors and outliers [36]. To avoid their confounding effects when simulating the trends of river conductivity and pH, this study averaged their original data by season for all sites firstly, defined as spring (March, April, May), summer (June, July, August), autumn (September, October, November) and winter (December, January, February) [13]. Moreover, for each river at the whole-basin scale, to calculate the correlation coefficients between the data of water quality and the factors more conveniently from vertical and horizontal perspectives later, this study calculated the annual average of pH, conductivity, agricultural and urban lands, the seasonal (winter and summer) average of pH, conductivity, NDVI, river discharge, the winter consumption of potential road salting, and the median level of these data over the past 20 years so that they were equal in quantity. The potential road salting is calculated as shown in Equations (1) to (3):

$$S_j = \frac{D_j R_j R_w S_u}{1000000}, 2000 \leq j \leq 2015 \tag{1}$$

$$D_j = length(find(T_{ji})), i = 12 \; Or \; 1 \le i \le 2 \tag{2}$$

$$T_{ji} = A_{ji} - \beta \le 0.5 \tag{3}$$

where $S_j$ is the potential road salting in the $j$ year (t); $D_j$ is the number of days to salt in the $j$ year (d); $R_j$ is the total length of roads in the basins in the $j$ year (m); $R_w$ is the average width of the major roads (m); $S_u$ is the usage of salting per square meter (g/m²); $length$ is a counting function; $find$ is a positioning function; $T_{ji}$ is the daily RST in the $i$ month of the $j$ year (°C); $A_{ji}$ is the daily air temperature in the $i$ month of the $j$ year (°C); $\beta$ is the empirical difference of air-to-road temperature (set $\beta = 4$).

*2.4. Water Quality Data Analysis*

For the seasonal averages of conductivity and pH for all sites, this study used the Theil-Sen regression and the Mann-Kendall test to identify their trends and statistical significances for each river at the whole-river scale, and for different seasons and reaches. The Theil-Sen regression is a non-parametric trend analysis that is able to resist the influence of outliers and provide robust slope estimates by calculating the median slope of paired data, but the slope will be greatly affected if there are many non-detects in the dataset [37,38] (Equation 4). However, this study solved this problem well by calculating the seasonal averages of conductivity and pH.

$$S_N = Median\left(\frac{N_j - N_i}{j - i}\right), \qquad 2000 \le i \le j \le 2021 \tag{4}$$

where $S_N$ is the estimate of trend slope in the data $N$; $N_j$ is the data $N$ in the $j$ year; $N_i$ is the data $N$ in the $i$ year. The $N$ has upward and downward trends when $S_N > 0$ and $S_N < 0$, respectively.

The Mann-Kendall test is a two-sided non-parametric trend test, which is normally used in conjunction with the Theil-Sen regression to obtain a full understanding of the trend. It is calculated as Equations (5) to (8):

$$S = \sum_{j=1}^{n-1} \sum_{i=j+1}^{n} sgn(N_j - N_i) \tag{5}$$

$$sgn(N_j - N_i) = \begin{cases} 1, N_j - N_i > 0 \\ 0, N_j - N_i = 0 \\ -1. N_j - N_i < 0 \end{cases} \tag{6}$$

$$s(S) = \frac{n(n-1)(2n+5)}{18} \tag{7}$$

$$Z = \begin{cases} \dfrac{S-1}{\sqrt{s(S)}}, S > 0 \\ 0, S = 0 \\ \dfrac{S+1}{\sqrt{s(S)}}, S < 0 \end{cases} \tag{8}$$

where $S$ is the Mann-Kendall statistic; $sgn$ is a step function; $N_j$ and $N_i$ are the data $N$ in the $j$ and $i$ year; $s(S)$ is the variance of $S$; $n$ is the length of time series; and $Z$ is the standardised test statistics. This study set the null hypothesis (H₀) and alternative hypothesis (H₁), meaning that the trend of data was insignificant and significant, respectively ($\alpha = 0.05$). When $|Z| \ge Z_{1-\alpha/2}$ ($Z_{1-\alpha/2} = 1.96$), the H₁ is accepted and recorded as 1, and the trend of data is significant (the greater the $|Z|$, the higher the significance) [39,40].

Apart from the trends of conductivity and pH, their extreme values are also helpful to understanding the status and hazards of salinisation and alkalinisation across each river. Therefore, this study made boxplots for the conductivity and pH of all sampling sites in each river to observe their extreme values more visually [41]. The aforementioned data pre-processing and analysis were processed by MATLAB R2021a.

*2.5. Correlation Coefficient Calculation*

Before calculating the correlation between the data of the water quality and the factors, a Shapiro–Wilk test was carried out for all variables ($\alpha = 0.05$) [42], which proved to be good at testing the normal distribution for small samples so that it accorded with this study. When the two variables conformed to the normal distribution, the Pearson's correlation coefficient ($\alpha = 0.05$) was chosen for them, which was widely used in the measurement of the degree of linear dependence between two variables [43]. Otherwise, Kendall's Tau correlation coefficient ($\alpha = 0.05$) was chosen, which was less influenced by the outliers and small samples and only used for small datasets influenced by outliers strongly [44]. The ranges of these two correlation coefficients are $-1 < r < 1$, and the two variables have linear positive, negative and no correlation when $r > 0$, $r < 0$ and $r = 0$, respectively. All correlation coefficients and tests were calculated by SPSS 26.0.

## 3. Results

*3.1. Trends of Conductivity and pH*

From the vertical perspective, the results for water quality data at the whole-basin scale show that the conductivities of the three rivers (River Tame, Trent, and Mersey) have significant downward trends with a median decrease of 0.06 to 0.11 mS/cm (Table 1, Figure 2), and the pH values of three of the four rivers (River Tame, Trent, and Avon) have significant upward trends with a median increase of 0.05 to 0.40 (Table 2, Figure 3), Furthermore, the conductivity and pH in each river also demonstrate different median and significant upward or downward trends in the specific river reaches and seasons, which can be called the regionality and seasonality of conductivity and pH. On the other hand, it is worth noting that the three rivers (River Tame, Trent, and Mersey) experienced extreme conductivity in 2010 and 2013.

**Table 1.** The test result ($|Z| \geq 1.96$) of conductivity trends for each river, each river in winter and summer, different reaches, and different reaches in winter and summer.

|  | River Tame | | River Trent | | River Mersey | | River Avon | |
|---|---|---|---|---|---|---|---|---|
|  | h | Z | h | Z | h | Z | h | Z |
| River | 1.00 | −3.14 | 1.00 | −2.85 | 1.00 | −2.35 | 0.00 | 0.03 |
| River (winter) | 1.00 | −3.01 | 1.00 | −3.12 | 1.00 | −2.10 | 0.00 | −0.97 |
| River (summer) | 0.00 | −0.54 | 0.00 | 0.20 | 0.00 | −1.15 | 1.00 | 2.57 |
| Upstream | 1.00 | −2.75 | 1.00 | −2.31 | 0.00 | −0.91 | 0.00 | −1.01 |
| Midstream | 1.00 | −3.93 | 0.00 | −0.41 | 0.00 | 0.18 | 0.00 | 0.93 |
| Downstream | 1.00 | −2.74 | 1.00 | −2.01 | 1.00 | −1.99 | 0.00 | 0.38 |
| Upstream (summer) | 0.00 | −1.25 | 0.00 | 0.02 | 0.00 | −0.34 | 0.00 | 0.27 |
| Upstream (winter) | 1.00 | −2.21 | 1.00 | −2.89 | 0.00 | −0.75 | 0.00 | 0.73 |
| Midstream (summer) | 0.00 | −1.15 | 0.00 | 1.02 | 0.00 | 0.31 | 0.00 | 1.52 |
| Midstream (winter) | 1.00 | −2.30 | 0.00 | −0.66 | 0.00 | 0.43 | 0.00 | 0.23 |
| Downstream (summer) | 0.00 | −1.17 | 0.00 | −1.10 | 0.00 | −1.55 | 0.00 | 1.75 |
| Downstream (winter) | 1.00 | −2.17 | 0.00 | −0.38 | 1.00 | −2.05 | 0.00 | −1.46 |

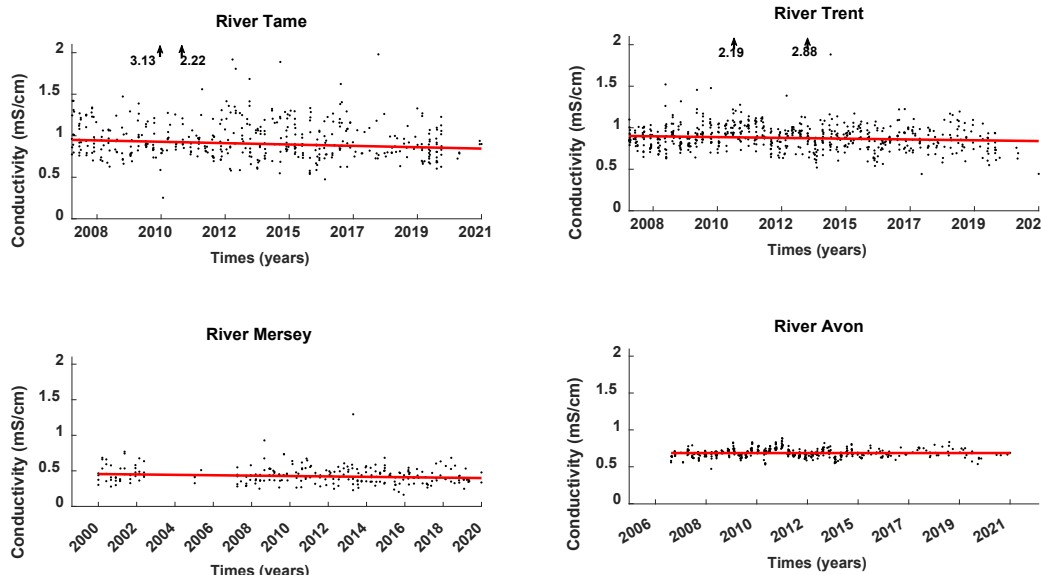

**Figure 2.** Conductivity of the whole rivers (The arrows represent outliers).

**Table 2.** The test result ($|Z| \geq 1.96$) of pH trends for each river, each river in winter and summer, different reaches, and different reaches in winter and summer.

| | River Tame | | River Trent | | River Mersey | | River Avon | |
|---|---|---|---|---|---|---|---|---|
| | **h** | **Z** | **h** | **Z** | **h** | **Z** | **h** | **Z** |
| River | 1.00 | 13.56 | 1.00 | 9.29 | 0.00 | −0.16 | 1.00 | 2.14 |
| River (winter) | 1.00 | 5.79 | 1.00 | 6.59 | 0.00 | 1.74 | 1.00 | 2.61 |
| River (summer) | 1.00 | 5.48 | 1.00 | 3.14 | 0.00 | −0.52 | 0.00 | −0.38 |
| Upstream | 1.00 | 7.62 | 1.00 | 6.14 | 0.00 | 1.50 | 0.00 | 1.15 |
| Midstream | 1.00 | 4.27 | 1.00 | 3.86 | 0.00 | 0.85 | 0.00 | 1.63 |
| Downstream | 1.00 | 7.16 | 1.00 | 2.32 | 1.00 | 2.27 | 0.00 | 1.24 |
| Upstream (summer) | 1.00 | 3.58 | 1.00 | 2.87 | 0.00 | 1.04 | 0.00 | −0.25 |
| Upstream (winter) | 1.00 | 2.77 | 1.00 | 3.02 | 0.00 | 1.88 | 0.00 | −0.53 |
| Midstream (summer) | 0.00 | 1.75 | 0.00 | 0.09 | 0.00 | 0.18 | 0.00 | 0.25 |
| Midstream (winter) | 0.00 | 1.71 | 1.00 | 2.69 | 0.00 | 0.22 | 1.00 | 2.33 |
| Downstream (summer) | 0.00 | 1.86 | 0.00 | 0.34 | 0.00 | 0.04 | 0.00 | 0.54 |
| Downstream (winter) | 1.00 | 3.84 | 1.00 | 2.24 | 1.00 | 2.38 | 0.00 | 1.50 |

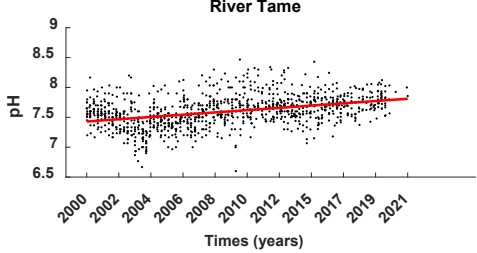

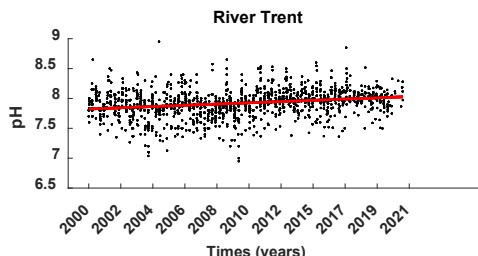

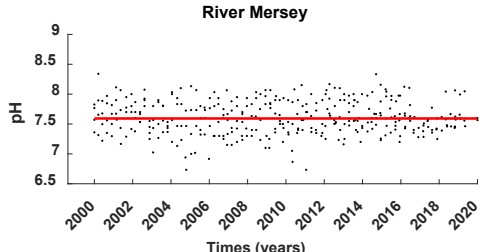
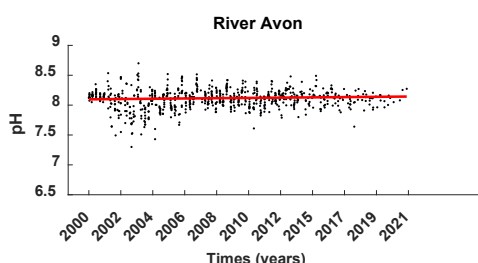

**Figure 3.** pH of the whole rivers.

### 3.1.1. The Regionality of Conductivity

When the rivers pass through the reaches with many anthropogenic disturbances, the conductivity usually changes greatly. The median of conductivity in the midstream of the River Tame flowing through the Lake Lea Marston is about 0.2 mS/cm lower than that in the upper reach near Birmingham (Figures 1 and 4), which is likely due to the purification of the lake [33] and the difference in urban lands between them. Furthermore, the conductivity of the River Mersey has the highest median level (0.45 to 0.55 mS/cm) and the most significant downward trend ($Z = -1.99$) in the lower reach (Figure 4, Table 1), which is explored later.

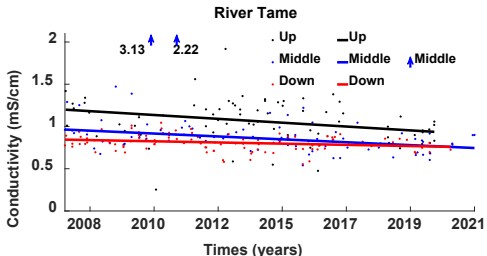
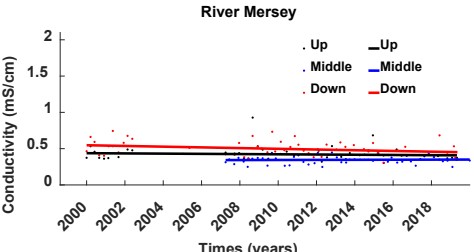

**Figure 4.** Examples of comparison of conductivity in the upper, middle, and lower reaches of the rivers.

### 3.1.2. The Seasonality of Conductivity

There are also some points worthy of attention about the seasonality of conductivity, especially in the River Tame and River Avon. Normally, the discharge of UK rivers in winter is much greater than that in summer [45], so the conductivity in summer is expected to be higher than that in winter. However, the conductivity of the River Tame in winter was higher than that in summer from 2008 to 2017 (Figure 5), which might indicate that there was more external input in winter. Different from the other three rivers, the conductivity of the River Avon in summer has a significant upward trend and its median in summer is higher than that in winter in all reaches (Figures 5 and 6), which is discussed later.

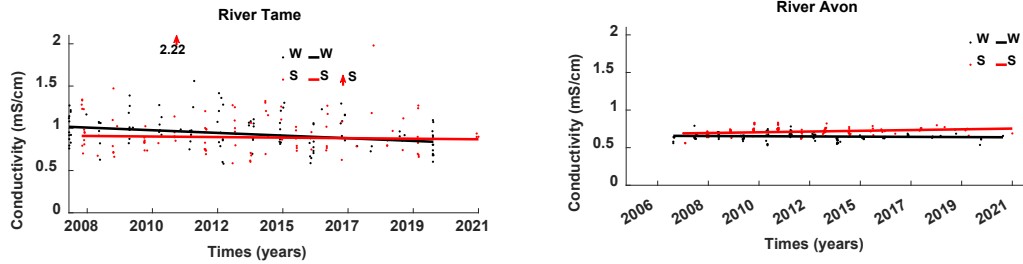

**Figure 5.** Examples of comparison of conductivity in summer and winter. (W: winter; S: summer).

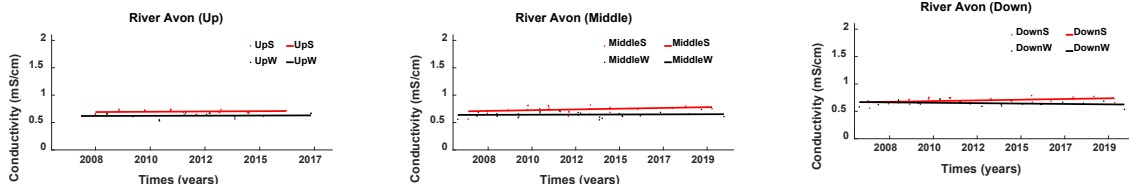

**Figure 6.** Comparison of conductivity in the upper, middle, and lower reaches of the river in winter and summer.

### 3.1.3. The Regionality of pH

Like conductivity, the pH of each river also has regionality. For instance, the median pH in the upstream River Tame is about 0.2 lower than that in the midstream (Figure 7); The median pH in the lower River Trent dominated by agricultural lands is about 0.1 higher than that of the middle reaches near Nottingham (Figures 1 and 7). Notably, the pH in the upstream River Trent has the most significant upward trend ($Z = 6.14$) (Table 2), and the median of it is only 0.02 different from that in the lower reaches in 2019 (Figure 7), which may reflect the impact of anthropogenic disturbance.

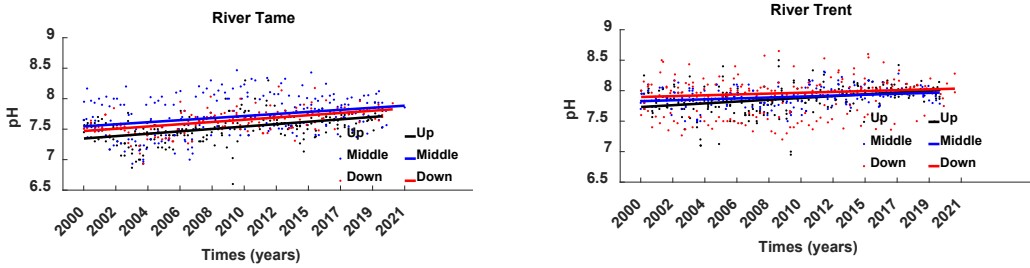

**Figure 7.** Examples of comparison of pH in the upper, middle, and lower river reaches.

### 3.1.4. The Seasonality of pH

Generally, in the UK summer, due to the higher water temperature, lower $SO_2$ in the air, and more vigorous photosynthesis of aquatic plants, the concentration of $CO_2$ and $SO_2$ in the rivers are lower than those in winter [46,47], resulting in the pH of the river in summer being higher than that in winter. Therefore, the information in Figure 8 is interesting—the pH of the River Trent conforms to this law, while that of the River Avon is contrary to it. However, not all reaches of the River Trent accord with the law, as the median pH in winter at the lower reach has been higher than that in summer since 2010 (Figure 9). In addition, the pH trends of both rivers in winter are more significant than in summer (Table 2). The causes for these phenomena are worthy of discussion later.

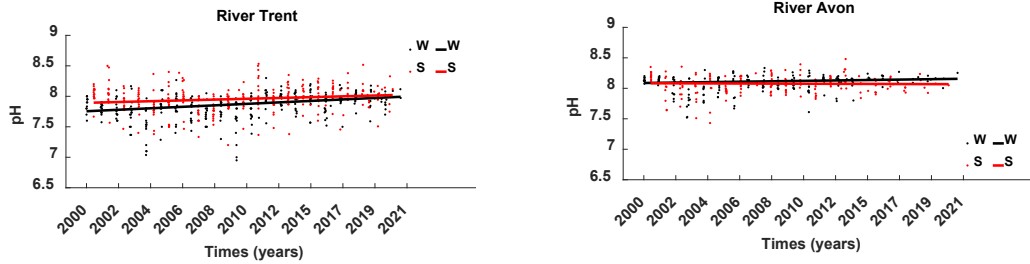

**Figure 8.** Examples of comparison of pH in summer and winter.

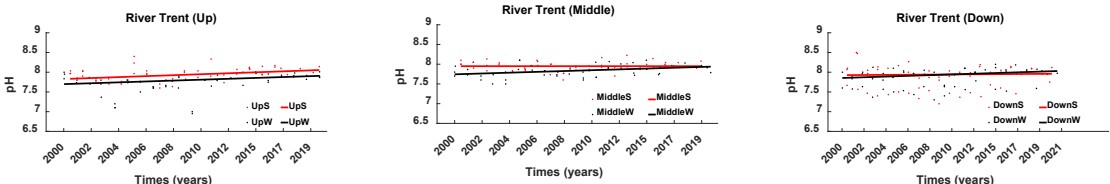

**Figure 9.** Examples of comparison of pH in the upper, middle, and lower reaches of the river in winter and summer.

### 3.2. Comparison of Conductivity and pH between All Rivers

From the horizontal point of view, there are some interesting points when comparing the conductivity and pH differences of the four rivers in this study. For the conductivity, the absolute value of conductivity trend significance in the three rivers is greater than that in the River Avon (Table 1). In addition, in Figure 10, the smallest fluctuation range (0.55–0.82 mS/cm) of conductivity is in the River Avon, while the largest fluctuation range is in the River Tame (0.48–1.47 mS/cm) likely due to different level of anthropogenic disturbances. Surprisingly, the River Mersey receiving external inputs from the Greater Manchester Canal [48] has the lowest conductivity level, the reasons for which are worth exploring later.

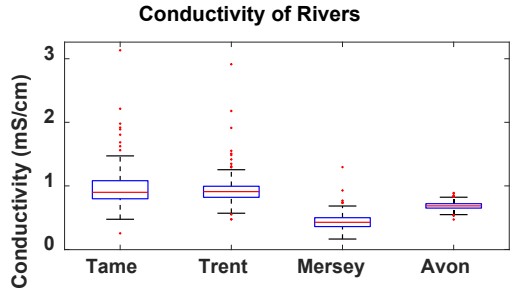

**Figure 10.** Differences in conductivity of the different rivers.

For the pH, the median pH in the three rivers (River Tame, Trent, and Mersey) are lower than that in the River Avon (Figure 11), which may indicate that the anthropogenic disturbance in the River Avon is the lowest. Moreover, although the medians of pH in the River Tame (7.60) and River Mersey (7.59) are extremely close, they have the largest and smallest trend significance (Table 2), respectively, which is worthy of discussion later.

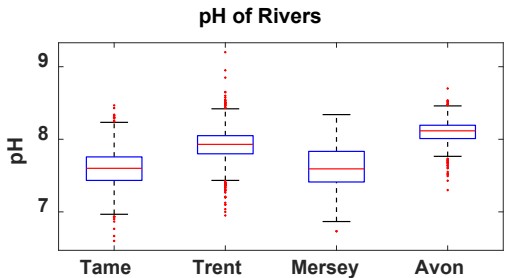

**Figure 11.** Differences in pH of the different rivers.

## 4. Discussion and Conclusions

### 4.1. Reasons for the Trends of River Conductivity

The difference between the results and the hypothesis comes from the salinisation trends. According to the positive correlation between conductivity and potential road salting (e.g., $r = 0.71$, $p = 0.03$) in Table 3, road salting is declining in the UK (but it is still one of the main external inputs of river salinity in winter, which may lead to higher medians and greater trend slopes of conductivity in winter than in summer). This is contrary to the trends shown in some other regions of the world, especially the USA (the main basis for the hypothesis). As mentioned before, road salting is related to climate and road mileage, so the reasons for this phenomenon may be due to: (1) The colder winter and more frequent snowstorms in the USA [49]; (2) The growth rate of road mileage in the USA (6.1%) [50] is much higher than that in the UK (2.1%) [51] over the past 20 years. Similarly, in the UK, compared with other rivers, the potential road salting of the River Mersey is the smallest due to the shortest major road mileage (107.9 km$^2$) in the basin, the small gap between the annual average number of days to salt (no more than 15 days), and the higher river discharge in winter (ranks second), which may be one of the reasons for the lowest median of conductivity in the River Mersey (Figure 10). In addition, the large amount of road salting caused by extreme cold weather in 2010 and 2013 [52] may be the main reason for the outliers of conductivity of some rivers in these two years (Figure 2), which is helpful to understand the effect of road salting on conductivity.

**Table 3.** The results of the correlation coefficients (the Kendall's Tau correlation coefficients are only calculated between the water quality data and landcover data) between the annual or seasonal average of the water quality and factors for each river in the past 20 years ('*' means significant).

| Water quality | Factors | Tame | | Trent | | Mersey | | Avon | |
|---|---|---|---|---|---|---|---|---|---|
| | | r | p | r | p | r | p | r | p |
| pH (annual) | Agricultural lands | 0.33 * | 0.03 * | 0.39 * | 0.02 * | 0.49 | 0.22 | 0.08 | 0.68 |
| | Urban lands | −0.33 | 0.11 | 0.01 | 0.97 | 0.13 | 0.52 | −0.08 | 0.71 |
| pH (summer) | Discharge (summer) | −0.01 | 0.96 | −0.47 | 0.07 | 0.46 | 0.07 | 0.38 | 0.15 |
| | NDVI (summer) | −0.15 | 0.53 | −0.08 | 0.74 | 0.13 | 0.60 | −0.12 | 0.61 |
| pH (winter) | Potential road salting (winter) | 0.18 | 0.51 | −0.18 | 0.51 | −0.18 | 0.51 | 0.15 | 0.57 |
| | Discharge (winter) | 0.08 | 0.76 | 0.22 | 0.41 | 0.26 | 0.33 | 0.04 | 0.88 |
| | NDVI (winter) | 0.16 | 0.48 | 0.42 | 0.06 | 0.04 | 0.86 | 0.35 | 0.12 |
| Conductivity (annual) | Agricultural lands | 0.21 | 0.42 | 0.21 | 0.37 | −0.25 | 0.25 | 0.07 | 0.79 |
| | Urban lands | −0.21 | 0.42 | 0.33 | 0.14 | −0.01 | 1.00 | 0.00 | 1.00 |
| Conductivity (summer) | Discharge (summer) | −0.59 | 0.12 | −0.56 | 0.11 | −0.50 | 0.10 | −0.86 * | 0.00 * |
| | NDVI (summer) | −0.39 | 0.17 | −0.37 | 0.24 | −0.48 | 0.06 | 0.10 | 0.75 |

| Conductivity (winter) | Potential road salting (winter) | 0.71 * | 0.03 * | 0.54 * | 0.04 * | 0.83 * | 0.03 * | 0.25 | 0.52 |
|---|---|---|---|---|---|---|---|---|---|
| | Discharge (winter) | 0.00 | 1.00 | −0.81 * | 0.02 * | −0.71 * | 0.01 * | −0.64 | 0.07 |
| | NDVI (winter) | −0.71 * | 0.01 * | −0.41 | 0.15 | −0.58 * | 0.02 * | −0.41 | 0.15 |

Furthermore, the results in Tables 3 and 4 also show that the river conductivity trends in the UK may be affected by river discharge, NDVI and urban lands. Specifically, the conductivity is negatively correlated with river discharge (e.g., $r = −0.81$, $p = 0.02$), and similar phenomena have been observed globally—less river discharge concentrates salt ions in the rivers [31], which may explain the significant rise of conductivity in the River Avon in the summer (Table 1, Figure 5).

**Table 4.** The results of the correlation coefficients between the median level (at a 20-year scale) of the water quality and factors in all rivers ('*' means significant).

| Water Quality | Factors | r | p |
|---|---|---|---|
| pH (annual) | Agricultural lands | 0.90 * | 0.04 * |
| | Urban lands | −0.34 | 0.66 |
| pH (summer) | Discharge (summer) | 0.29 | 0.71 |
| | NDVI (summer) | 0.72 | 0.28 |
| pH (winter) | Potential road salting (winter) | 0.01 | 0.99 |
| | Discharge (winter) | 0.31 | 0.69 |
| | NDVI (winter) | 0.99 * | 0.01 * |
| Conductivity (annual) | Agricultural lands | 0.53 | 0.47 |
| | Urban lands | 0.84 * | 0.04 * |
| Conductivity (summer) | Discharge (summer) | 0.41 | 0.59 |
| | NDVI (summer) | −0.33 | 0.67 |
| Conductivity (winter) | Potential road salting (winter) | 0.95 * | 0.04 * |
| | Discharge (winter) | 0.37 | 0.63 |
| | NDVI (winter) | 0.03 | 0.97 |

At the same time, the negative correlation between NDVI and conductivity has been observed in the Mississippi River [53] and many coastal areas [54], usually because higher soil and river conductivity may damage the health of plants (decrease of NDVI), which seems to be applicable to the results in Table 3 for the River Tame ($r = −0.71$, $p = 0.01$) and River Mersey ($r = −0.58$, $p = 0.02$).

Interestingly, while this study does not find a significant correlation between conductivity and urban lands on the vertical comparison, it is observed on the horizontal comparison ($r = 0.84$, $p = 0.04$), which is consistent with Carpenter's [55] finding that rivers receive more salt as they flow through cities. In addition, the results of the horizontal comparison may be used to account for the differences in conductivity between different reaches [12], for instance, the conductivity in the upper reaches of the River Tame flowing through Birmingham (more urban lands) is much higher than that in the middle and lower reaches dominated by agricultural lands (Figure 4). However, there is not a significant correlation between conductivity and agricultural lands, which is different from the phenomena observed in the central USA, whose causes include direct input of salt ions from fertilisers, and enhanced nitrification of fertilisers leading to the acceleration of the soil ion exchange rate [13].

Of course, the results in Tables 3 and 4 do not explain the causes for the trends of river conductivity in the UK comprehensively. In fact, the treatment measures may also play a key role in influencing the trends of conductivity. For example, since the 1980s, as a result of a series of legislative changes, the pollutants discharged into the upstream River Tame have decreased, and the closure of some plants and construction of the Lake lea

Marston in the middle reach has led to the deeper ameliorating of the water quality [56], which is consistent with the most significant downward trend ($Z = -3.93$) of conductivity in the midstream of the River Tame (Table 1). Similarly, the River Mersey, one of the most polluted rivers in the UK's history [34], has had a great ameliorating of the river aquatic ecosystem since the Mersey Basin Campaign in 1985 [57], which is dedicated to river cleaning and may be one of the main reasons for the significant downward trend in the lower River Mersey (Table 1).

*4.2. Reasons for the Trends of River pH*

Current studies indicate that agricultural activities can accelerate rock weathering and base actions leaching, resulting in the alkalinisation of rivers [27,28], which seems to explain the positive correlation between pH and agricultural lands in the River Tame and the River Trent (Table 3). On the other hand, the results in Table 4 show a significant positive correlation between pH and NDVI in winter ($r = 0.99$, $p = 0.01$), which is consistent with the phenomenon that the growth of NDVI usually means the improvement of water quality, especially in the acidified rivers [53,58]. Unfortunately, this study does not observe the significant correlations between pH and urban lands, river discharge and road salting, which are usually considered to be the factors affecting river acidification [13,59–61]. The reasons for this may include: (1) The period of the observed data is probably insufficient; (2) The correlation may be nonlinear [12].

In addition to the aforementioned reasons, some factors are also considered to have an impact on the river pH, such as acidic deposition, geological conditions and treatment measures. For the UK, acidic deposition is one of the reasons for river acidification, particularly in the regions with serious industrial pollution, which may explain the phenomena that the River Mersey and River Tame, near heavy industrial cities (e.g., Manchester and Birmingham), have a lower median pH (Figures 1 and 11). Also, acidic deposition in the UK has seasonality, the reasons for which include: (1) The rate of heating supply in winter is much higher than the utilisation rate of air conditioners in summer, resulting in higher sulfur emissions in winter than in summer; (2) There are lower temperature and more precipitation in winter than in summer, causing acidic deposition in winter more easily [62]. Fortunately, since the last century, with the promulgation of the Clean Air Act and the promotion of clean energy [63,64], the acidic deposition in the UK has decreased significantly, especially in winter [65], which is consistent with the more significant upward trends of pH in winter for each river (Table 2). However, even though the decrease in acidic deposition may cause the increase in river pH, it seems unlikely that river pH is higher in winter than that in summer due to differences in the solubility of $CO_2$ and $SO_2$ in the river between winter and summer [46,47], unless there are additional alkaline inputs or extreme hydrology events. In fact, there is a large amount of limestone in the middle and lower reaches of the River Trent and the middle and upper reaches of the River Avon. This geological structure usually makes alkaline substances enter the river with surface runoff under the influence of chemical weathering [66], which may be one of the reasons for the higher pH in winter in these rivers (Figures 8 and 10). In addition, some local treatments may also play a key role in mitigating river acidification. For example, the joint committee composed of local government and industry in Manchester has issued a series of bills to ameliorate the water quality in the downstream river [57], which may be one of the main reasons for the rise of pH there over the past 20 years (Table 2). Similarly, the treatments of the upper and middle River Tame mentioned above [56] may also apply here. Considering that there are a lot of same anthropogenic disturbances in the upper and middle reaches of the River Tame and River Mersey (e.g., a high level of urbanisation and industrialisation, and the impact of the Manchester Canal [34]) (Figure 1), while the treatments of a single reach usually have a limited impact on the water quality at the whole-basin scale, which may explain the phenomenon that the pH of the River Tame has a significant upward trend at the whole basin scale,    the River Mersey has no such trend (Table 2).

*4.3. Potential Effects of Extreme River Conductivity and pH*

Although the conductivity and pH for each river in this study are at 'Good' status (6 < pH < 9; annual average conductivity < 1 mS/cm) for most of the time according to the Water Framework Directive (Standards and Classification) Directions (England and Wales) 2015, some disturbing extremes are still observed. For example, the conductivity of the River Tame was as high as about 3.13 mS/cm in 2010 (Figures 2 and 10), and the extremes of pH in the River Trent in 2006 and the River Mersey in 2003 were 9.2 and 6.7, respectively (Figures 3 and 11). In general, it is necessary to pay attention to these extreme values, as they may imply damage to the river aquatic ecosystem. Some studies have shown that high river conductivity will compress the living space of aquatic organisms (e.g., invertebrates) and coastal plants [67,68], and too low a pH may also lead to the collapse of aquatic ecosystems [69]. In addition, the toxicity of the base cations is also different at different pH. For instance, the toxicity of $NH_4^+$ increases with pH exceeding 8 [70], while that of $Zn^{2+}$ increases with pH drops below 8 [71]. Notably, in populated areas, high river conductivity may cause the leaching of metal ions from old pipes, and threaten the safety of drinking water [4], which may occur in the River Tame basin. Thus, for the UK, it is important to continue to strengthen the management of acid inputs and salt inputs, regulate the use of fertilisers and easily weathered materials, and control the area of agricultural land.

*4.4. Limitations and Recommendations*

The main limitations of this study come from the observation data, including (1) An insufficient period: Generally speaking, to reflect a long-term trend of data better, the length of data recording should be at least 25 years [12]. However, the water quality database of the Environment Agency and the UK Centre for Ecology and Hydrology cannot meet this requirement, which may cause some data to fail to show significant trends and correlations; (2) Fewer types: Limited by the dataset, this study only uses conductivity and pH to represent the river salinisation and alkalinisation, respectively. It is worth mentioning that there is not even an accurate record for road salting per year, which may cause uncertainties in this study; (3) Measurement uncertainties and errors: Although the seasonal averages are used to reduce the impact of uncertainties and errors on trends analysis in this study, it can not be avoided totally due to the monthly and irregular sampling period. (4) Policy changes: In the past 20 years, the UK's water quality monitoring standards have undergone three reforms, which caused a long-term gap in some monitor sites, which may cause uncertainties in the trends obtained in this study. Furthermore, this study only selected the data of four representative rivers for analysis, which are distributed in the central and western parts of England with anthropogenic disturbance factors. Therefore, the results of this study may not be fully representative of all UK rivers, especially the natural rivers with less anthropogenic disturbances.

Based on the aforementioned reasons, this study puts forward some suggestions: (1) To establish a comprehensive UK river water quality dataset as soon as possible; (2) Due to the limited data, this study does not discuss the degree to which the river is affected by acidic input and alkaline input. Future research may be able to make a more comprehensive quantitative analysis from the perspective of atmospheric emissions and changes in base cations; (3) This study does not explore the specific mechanism of the impact of river salinisation and alkalinisation on the environment, such as the chemical cycle of toxic ions in the process of salinisation. To better understand the hazards of salinisation and alkalinisation, this may be a useful direction; (4) Although most rivers in the UK are affected by anthropogenic disturbance (the level of urbanisation in the UK is very high), which means that the results of this study may be applied to more rivers in the UK, there are still some natural rivers that have not been considered. Future research can analyse the salinisation and alkalinisation trends of UK rivers on a broader scale; (5) As the salinisation and alkalinisation of rivers are mainly affected by anthropogenic

disturbance and climate change, the results of this study may be applicable to the rivers of some European countries with temperate climate and high urbanisation. However, for countries with large areas and diverse climates such as the USA or China, the trends may be complex and different and deserve further exploration.

**Author Contributions:** Conceptualization, S.J. and D.H.; Methodology, S.J., D.H., Q.W., X.W. and S.D.; Writing—original draft, S.J. and X.W.; Writing—review and editing, D.H., Q.W. and S.D. All authors have read and agreed to the published version of the manuscript.

**Funding:** This publication has been funded by Dawei Han, a member of *Water* Editorial Board with one free publication.

**Data Availability Statement:** Not applicable.

**Acknowledgments:** The authors would like to thank the support provided by Jie Shi.

**Conflicts of Interest:** The authors declare that they have no conflict of interest.

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
