# Peer review of "Are UK Rivers Getting Saltier and More Alkaline?"

_water, doi:10.3390/w14182813_

Round 1

Reviewer 1 Report

This is a good manuscript. The work was designed well, the presentation of results and discussion were well organized. I have few comments as follows.

-          The topic is important, but the authors need to point out their objectives clearly and novelty of the study, which need to be stressed in the revised version.

-          Why the authors use Theil-Sen regression and Mann-Kendall test, what’s the advantages and disadvantages of those tests.

Author Response

Thanks for your helpful comments. Please see the attachment.

Reviewer 2 Report

Review of “Are UK rivers getting saltier and more alkaline?” by Jiang et al. (Water-1881190).

The authors using data mining examine trends in four UK rivers of pH and conductivity.  They explain their criteria using various statically methods were strict and only the four rivers discussed could evaluated over a period of 20+ years.  This is surprising to me because as a geochemist I never trust pH meters; I prefer to calculate the pH from the carbonate system.  I have numerous major and a few minor comments.  Although the writing is fine, there are too many other problems that the authors must address.  Therefore, I recommend that this manuscript be rejected for publication in Water.  If the authors decide to revise it, it should be treated as a new submission.

Major Comments

1)     My most serious comment is I do not know why this manuscript was written and submitted for publication. It seems to better suited as technical note published by the EAWQDA.

2)     Much more information is needed for each watershed.  During the 20+ year record for which the authors have chemistry records a) did the population changed, b) how did the land use change, c) did the type of industries change, d) did the climate change, and e) did the hydrology change (i.e., did sections of the river change from gaining to losing zones, did the channel morphology change, were new infrastructure built)?

3)     Most of the figured need to be redrafted.  As I often find when reviewing manuscripts for Water, the font size used is too small and must be enlarged.  Figure 1: The maps of the UK in the upper left corner are very difficult to read. I expect that they show the location of each river.  With exception of River Trent, I had a hard time finding the location of the other three rivers especially River Tame and River Avon.  I suggest including a box for the area shown in the main panels.  Figures 2, 4, 5, 6, and 10: Please change the limits and units of the y-axis to, respectively, ‘0 – 2’ and ‘mS/cm’. There are only a few points greater then 2 mS/cm and these could be included by using vertically-up arrows placed at the appropriate time with the value written beneath.

4)     Please change the unit of conductance from µS/cm to mS/cm throughout the manuscript. 

Minor Comments

Line 53 and elsewhere in the manuscript: Please use the subscript and superscript when referring to ions and units; ‘Na+’ is not the same as ‘Na+’ nor is ‘km2’ the same as ‘km2’.

Lines 225, 227, 228, 230, 244, 257, 259, 270, 282, and 284: Please change ‘Time(years)’ to ‘Times (years)’.

Line 239: How does ‘purification of the lake’ lower the conductance of the water?  Reference #39 is nearly impossible to find and because it is not a journal article its conclusions are questionable.

Line 325: Please change ‘condenses the concentration of’ with ‘concentrates’.

Line 329” Please change ‘River Mississippi’ to ‘Mississippi River’.

Line 343 and elsewhere in text: ‘agricultural fertilisers (sic)’ seems redundant.  Please change to ‘fertilizers’.

Line 374: Please change ‘rive’ to ‘river’.

Line 409: Significant figures.  Please change ‘3132 µm/cm’ to ‘about 3.15 mS/cm’.

Author Response

(The authors gave the same response as above.)

Reviewer 3 Report

Recommendation: Accept in present form

Manuscript Number: water-1881190

Are UK Rivers Getting Saltier and More Alkaline?

Shan Jiang, Xuan Wu, Sichan Du, Qin Wang, Dawei Han

Overview and general recommendation:

Thank you for the opportunity to review the manuscript "Are UK Rivers Getting Saltier and More Alkaline?" for the Water journal. Given that the topic of your manuscript centers on the rivers, I believe that the subject matter is in line with the scope of Water journal.

The work deals with an interesting topic and seems to be very current. Research on rivers and their monitoring and protection is necessary, especially in human-altered catchments. Rivers are an important source for humans to obtain freshwater and have a significant impact on human survival and socio-economic development. With the accelerated climate change and aggravating anthropogenic influences, it is important to identify the trends and causes of river salinisation and alkalization. This study has focused on the UK rivers. To reflect the trends of river salinisation and alkalisation in different parts of the UK and explore their causes, this study took into account many factors. The results of the research are not only of scientific but also practical importance and can be used in the development of water management strategies in this country.

The paper is well-organized, containing all of the expected components. In the Introduction, the authors discussed the causes of river pollution and the related problems, focusing mainly on the alkalisation and salinisation of rivers. The proposed methodology is professional and effective in attaining the object of this work. The research results described in the Discussion and Conclusions section are presented clearly and comprehensively. The tables and figures are also adequate to the obtained test results.

Author Response

Thank you so much for taking the time to read this manuscript, and your affirmation of the topic, structure and results of our manuscript, which is a great encouragement to us.

Reviewer 4 Report

The manuscript presents a very ambitious study to find some trends and causes of Rivers salinisation and alkalization in UK. Two indicators (conductivity and pH) were considered to evaluate the trends of salinisation and alkalization.

Even though a lot of work was carried out to find trends in pH and conductivity values over the time using modern chemometric approaches there are many limitations in this study. Maybe for this reason, the manuscript title is a question. In my opinion, is inappropriate to have a general conclusion for the UK Rivers if only four rivers were considered in the study.

Measurement uncertainties of the analytical methods were not considered when the statistical evaluation of the results was carried out. Are the results significantly different over the time, if consider the measurement errors? Other limitations are presented even by authors, the most important in my opinion the use of only pH and conductivity to evaluate this trend. In these conditions is difficult to have a rigorous conclusion if UK Rivers really getting saltier and more alkaline.

Author Response

(The authors gave the same response as above.)

Reviewer 5 Report

In this paper, Shan Jiang et al examined whether or not UK rivers getting saltier and more alkaline. Shan Jiang et al reached one major point "The results show that the UK rivers are becoming more alkaline with a median pH increase of 0.05 to 0.4". This paper is very interesting and has a great scientific impact. I suggest this paper be published after several modifications. Below are my suggestions:

Major suggestions:

(1) The authors may shorten the introduction part to make it more concise and clear. There may be too many words in the introduction part.

(2) We can see clear pH increasing for River Tame (Figure 3, 2000 to 2021). However, we cannot see this pH increasing for River Mersey (Figure 3, 2000 to 2021). The authors may deeply discuss what is the major factors that contribute to this difference.

(3) The authors found this trend in the UK. The authors may briefly discuss the possible impacts on other countries' rivers all over the world: if a similar study is performed in other countries, are the results similar to the UK? Why?

Minor suggestions:

(1) Line 120 Figure 1: Make the font size consistent with "Bristol" versus "Bath" versus "Manchester" versus "Derby".

(2) Line 151 16g/m2 versus Line 171 g/m2: please be careful with the number and the unit. Confirm the space between all numbers and all units. Check all the superscripts and subscripts.

(3) Line 225 to Line 227 Figure 2: A lot of outliner points. Please explain if possible.

(4) Line 309 UK versus Line 312 U.S. versus Line 315 U.S: Please make the writing consistent. 

(5) Line 345 to Line 347 Table 3: Please double-check the numbers. Also, check the digits of the number. 

Author Response

(The authors gave the same response as above.)

Round 2

Reviewer 4 Report

The authors responded to my comments and improved the manuscript.